# Mental Health during the COVID-19 Lockdown over the Christmas Period in Austria and the Effects of Sociodemographic and Lifestyle Factors

**DOI:** 10.3390/ijerph18073679

**Published:** 2021-04-01

**Authors:** Rachel Dale, Sanja Budimir, Thomas Probst, Peter Stippl, Christoph Pieh

**Affiliations:** 1Department for Psychotherapy and Biopsychosocial Health, Danube University Krems, 3500 Krems, Austria; rachel.dale@donau-uni.ac.at (R.D.); sanja.budimir@donau-uni.ac.at (S.B.); thomas.probst@donau-uni.ac.at (T.P.); 2Department of Work, Organization and Society, Ghent University, Sint-Pietersnieuwstraat, 9000 Ghent, Belgium; 3Austrian Federal Association for Psychotherapy (ÖBVP), 1030 Vienna, Austria; oebvp.stippl@psychotherapie.at

**Keywords:** mental health, COVID-19, Austria

## Abstract

Since the beginning of the COVID-19 pandemic a decline in mental health has been reported. This online study investigated mental health and well-being in Austria during a strict lockdown. In total, *N* = 1505 participants were recruited between 23 December 2020 and 4 January 2021 and levels of depression (PHQ-9), anxiety (GAD-7), sleep quality (ISI), well-being (WHO-5), quality of life (WHO-QOL) and stress (PSS-10) were measured. 26% scored above the cut-off for moderate depressive symptoms (PHQ-9 ≥ 10; ♀ = 32%; ♂ = 21%), 23% above the cut-off for moderate anxiety (GAF-7 ≥ 10; ♀ = 29%; ♂ = 17%) and 18% above the cut-off for moderate insomnia (ISI ≥ 15; ♀ = 21%; ♂ = 16%). Mean-scores for quality of life (psychological WHO-QOL) were 68.89, for well-being (WHO-5) 14.34, and for stress (PSS-10) 16.42. The youngest age group (18–24) was most burdened and showed significantly more mental health symptoms compared with the oldest age group (65+) in depressive symptoms (50% vs. 12%), anxiety symptoms (35% vs. 10%), and insomnia (25% vs. 11%, all *p*-values < 0.05). Mental health decreased compared to both the first lockdown earlier in 2020 and pre-pandemic data. Further analyses indicate these findings were especially apparent for the under 24-year-olds, women, single/separated people, low incomes and those who do not partake in any physical activity (all *p*-values < 0.05). We highlight the need for ongoing mental health support, particularly to the most burdened groups.

## 1. Introduction

The Coronavirus disease 2019 (COVID-19) pandemic has spread quickly around the world [1]. In Europe alone, during the survey period, there were over 28.1 million confirmed cases and over 597,000 confirmed deaths (as of 4 January 2021). Within Austria there were 364,600 confirmed cases and over 6300 confirmed deaths (as of 4 January 2021, [2]. COVID-19 spreads between people in contact [1] and therefore many governments have placed necessary restrictions to reduce contact between people, resulting in drastic changes to our lifestyles.

A number of studies have already shown that the concerns over the pandemic and the restrictions have had clear and large effects on mental health worldwide [3,4,5,6,7], and may lead to long-term impacts [8]. For example, a large meta-analysis found a prevalence of 34% for depression, 32% for anxiety and 30% for stress [9]. It is difficult to tease apart the specific reasons for this decline in mental health over the pandemic but fear of illness/death [10], economic problems/unemployment [11] and reduced social contact/isolation [12] have all been suggested as important factors. Therefore, it is important to identify sociodemographic and lifestyle-related risk factors to mental health during the pandemic.

Indeed, findings to date have already identified some crucial factors. Younger people have shown to have poorer mental health during the pandemic in Italy [13], Japan [14], Spain [15], India [16] and Iran [17]. Women were also more affected than men in the studies from Italy [13], Iran [17] and China [18]. A study from Portugal [19] found having work and being more physically active protect from mental health problems. The study from Japan [14] also found unemployment to be a risk factor. Results from a survey of a representative sample of the Austrian population during the first lockdown in spring 2020 indicate a similar pattern to the aforementioned findings, with younger adults (<35 years), women, people without work and lower incomes being particularly affected in their mental health [6]. Additionally, loneliness and perceived stress combined in the first lockdown have been shown to be risk factors for subsequent depression after the lockdown in Austria [20]. No changes in mental health burden were observed either between the first lockdown in spring 2020 and summer 2020 [21] or between the first lockdown in spring 2020 and autumn 2020 [22] in Austria.

Austria had a first lockdown in spring of 2020 (16 March 2020–30 April 2020, hereafter referred to as April lockdown), but over the summer the number of daily cases reduced and the restrictions were eased. However, in September 2020 the cases in Austria began to rise again and on 3 November 2020 the government imposed another lockdown which lasted in its strictest form (see below) until 8 February 2021 (hereafter referred to as December/January lockdown). As in the April lockdown, only essential business could remain open (e.g., supermarkets and pharmacies) and there were only five exceptions to the ban on entering public places; activities to avert an immediate danger to life, limb, or property; professional activity (if home-office is not possible); errands to cover necessary basic needs; care and assistance for people in need of support; exercise outdoors alone and with pets/people living in the same household. A distance of at least 1 m to other people had to be ensured. In Austria people were allowed to meet in groups of up to 10 people on the 24 and 25 December 2020 but no group gatherings were allowed at other times, including on the 31 December 2020.

In addition to the burden of another lockdown, this one took place over the Christmas holiday period. In countries such as Austria, Christmas is normally a significant social ritual where getting together in a group is important, for example, for reducing loneliness in elderly people [23], and a high participation in ritualised family celebrations has been shown to increase well-being and life satisfaction [24]. As such, the inability for some to partake in such rituals could be detrimental to the already decreased mental health of the general population. Furthermore, although the Christmas holiday period is a positive time for many, it can also be a stressor. Conflict experienced over the Christmas holidays increases negative affect [24] and the holidays are often a challenging time for those vulnerable to mental health issues. For example, annual suicide rates peak on New Year’s Day in the USA [25] and Austria [26]. Additionally, Christmas can be associated with reduced life satisfaction and emotional well-being [27], lower mood [28] and higher levels of loneliness [29]. Therefore, a lockdown over this period may be of particular concern for public mental health.

Given the previously observed mental health deterioration during the pandemic, it is important to monitor the mental health status of the population over the course of the pandemic, and particularly during important cultural and social occasions such as the Christmas holidays. As such, we measured mental health in Austria in the holiday period (23 December 2020–4 January 2021) with the aim of assessing how a lockdown over the Christmas holiday period has affected mental health in the Austrian population. Additionally, we consider which groups have been most affected by looking separately at gender, age, marital status, income and physical activity. We then put these results into context by comparing prevalence of moderate-severe depression, anxiety and insomnia symptoms with those from the first lockdown in spring of 2020 as well as with pre-pandemic data from Austria collected in 2018–2019 [30].

## 2. Materials and Methods

### 2.1. Study Design

An online survey was performed with Qualtrics^®^ (Qualtrics, Provo, UT, USA) [31] to measure mental health during the COVID-19 restrictions over the Christmas period in Austria. The survey started on the 23 December 2020 and ended on the 4 January 2021.

### 2.2. Study Sample

Austria has a total population of 8,894,380 (as of 31 October 2019, [32]). A representative sample according to age, gender, education, and region for Austria was recruited through a Qualtrics panel, who organised recruitment and data collection. We aimed for a representative sample size according to age, gender, education, and region, with age and gender quotas interlocked, of at least 1500 participants. Due to the restricted time period of the survey, not all quotas were fulfilled, for example young males and those with a high education level (see Appendix A). Demographic characteristics of the study sample (*N* = 1505) are presented in Table 1.

### 2.3. Measures

The measures were selected because they have been validated in German and are often used in the research literature to assess mental health and psychological symptoms, including in previous COVID-19 mental health studies (e.g., Pieh et al. [6]), thus increasing comparability.

### 2.4. Quality of Life (WHO-QOL BREF)

The WHOQOL-BREF [33] is an oft used measure of quality-of-life. The questionnaire is 26-item self-rating measure of physical health, psychological health, social relationships, and environment over the last two weeks. The WHOQOL-BREF is validated and has good to excellent psychometric properties of reliability [34]. The current study on used only the Psychological domain, which has a typical norm value of 71.5 [35]. Cronbach’s alpha for the psychological domain was α = 0.93 in the current sample.

### 2.5. Well-Being (WHO-5)

The WHO-5 questionnaire [36] measures well-being with five self-rating items rated on six-point Likert scales. The score can range from 0 (absence of well-being) to 25 (maximal well-being). The WHO-5 has good psychometric properties [37]. Cronbach’s alpha was α = 0.91 in the current sample.

### 2.6. Perceived Stress (PSS-10)

In the PSS-10 participants are asked to rate their stress-levels over the last month in a 10-item questionnaire using five-point scales ranging from 0 to 4. The PSS-10 is a reliable and valid measure of stress [38]. Cronbach’s alpha was α = 0.88 in the current sample.

### 2.7. Depressive Symptoms (PHQ-9)

Depressive symptoms were measured with the depression module of the Patient Health Questionnaire; the PHQ-9 [39], which contains nine self-rating items on a four-point scale, from 0 to 3. Additionally, PHQ-8 scores were calculated to compare the present depression levels with the pre-pandemic data from the Austrian Health Survey 2019 [30]. Cut-off points are 5 for mild, 10 for moderate and at least 15 for severe levels of depression [40]. The 10-point cut-off was used in the present study to define clinically relevant depression for both PHQ-8 and PHQ-9. Cronbach’s alpha was α = 0.9 for PHQ-9 and PHQ-8 in the current sample.

### 2.8. Anxiety (GAD-7)

The Generalized Anxiety Disorder 7 scale (GAD-7) [41] is a validated [42] measure of anxiety symptoms, with seven self-rating items on a four-point scale, from 0 to 3. Cut-off points are 5 for mild, 10 for moderate and 15 for severe anxiety symptoms. The 10-point cut-off was used in the current study to define clinically relevant anxiety. Cronbach’s alpha was α = 0.92 in the current sample.

### 2.9. Sleep Quality (ISI)

Insomnia levels and sleep quality were measured using the Insomnia Severity Index (ISI), which is a self-reported seven-item questionnaire, with each item on a four-point scale (from 0 to 4). The total score is categorised as follows: (1) no clinically significant insomnia (<7 points), (2) sub-threshold insomnia (8–14 points), (3) clinical insomnia (moderate severity) (15–21 points), and (4) clinical insomnia (severe) (22–28 points). The 15-point cut-off score was used in the present study to define clinically relevant insomnia. Cronbach’s alpha was α = 0.86 in the current sample.

### 2.10. Other Variables

Gender was coded as male/female, no participants identified as diverse. Age was coded as years, in the following categories: 18–24, 25–34, 35–44, 45–54, 55–64, 65+. Marital status was coded categorically as single, separated, divorced, cohabiting, married or widowed. Income was recorded as net monthly income in euros. Physical activity was the number of days per week in which participants engaged in at least 30 min of physical activity.

### 2.11. Statistical Analyses

Data were analysed using SPSS version 27 (IBM, Armonk, NY, USA) [43] and R version 4.0.3 (R Foundation for statistical computing, Vienna, Austria) [44]. Descriptive statistics describe the demographic characteristics of this representative Austrian sample (Table 1), mean values of each measure, and the percentage of each sub-group (gender, age, marital status, income and physical activity) showing above cut-off symptoms for depression, anxiety and insomnia (Tables 2–6). Chi-squared tests were performed to investigate differences between the above and below cut-off groups for each sub-group. ANOVAs (mental health scales as dependent variables and age group, marital status, net income or physical activity as between-subject variable), and Bonferroni-corrected post hoc tests were performed to compare the impact on mental health in different age, marital status, net income, and physical activity groups. *T*-tests were calculated to compare differences in the mental health measures between genders. As the effect size measure, η^2^ was used to categorize small (η^2^ = 0.01), medium (η^2^ = 0.06), and large (η^2^ = 0.14) effects for ANOVAs and Hedge’s g was calculated for gender (small effect: 0.2 to 0.5, medium effect: 0.5 to 0.8, large effect: >0.8 [45]).

An ANOVA was conducted to compare the mean values for depression and quality of life (PHQ-8 and WHOQOL-BREF scores) for the period before the pandemic in Austria (ATHIS data collection 2018–2019) [30], during the April lockdown in Austria [6] and the December/January lockdown (the current sample) and a Chi-squared test was conducted to compare the number of people over the PHQ-8 cut-off in these different samples. The ATHIS 2019 sample consisted of 15,461 participants but was weighted to produce a statistical sample of 7.4 million, representing the Austrian population of over 15-year-olds [30]. Finally, linear models were run in R to analyse the effect of lockdown on the likelihood of being over the cut-off for depression, anxiety or insomnia according to gender, age group and marital status. *p*-values of less than 0.05 were considered statistically significant (two-sided tests).

## 3. Results

The number of people scoring above the cut-offs for clinically relevant depressive, anxiety and insomnia symptoms for the total sample, and divided by gender are presented in Table 2. Mean scores for the measures of depression, anxiety, insomnia, quality of life, well-being and stress can also be seen here. In sum, over 26% of people scored above the cut-off for depression, over 23% scored above the cut-off for anxiety, and more than 18% above the cut-off for insomnia. Women showed poorer mental health than men on all measures (all *p* < 0.05).

Mental health differed significantly between age groups on all measures (all *p* < 0.01), with less impact as age increases. Those under 24 years of age are particularly impacted with a staggering 50% showing clinically relevant symptoms of depression and over 30% showing moderate anxiety. Those over 65 were less burdened than other groups. The means, standard deviations and the differences in mental health according to *age* category are presented in Table 3. For this, and each of the following analyses, significant post-hoc results can be seen in the Appendix A.

Results showing the means, standard deviations and the differences in mental health according to marital status are presented in Table 4. In sum, mental well-being differed according to marital status on all measured scales (all *p* < 0.05). Single and separated people were particularly burdened, for example 35% of single and 46% of separated people scored above the cut-off for moderate depressive symptoms. Married people reported better mental health than these groups.

The results for mental health according to income are presented in Table 5. To summarise, mental health improves with income across all measures (all *p* < 0.01), with those earning below €1000 per month net showing poorer mental health than other groups. Furthermore, physical exercise had a significant positive effect on mental health on all scales (*p* < 0.05). Those who do at least 30 min of physical activity at least once per week report better mental health than those who never do a minimum of 30 min exercise. The results are presented in Table 6.

In comparison with other representative surveys conducted in 2018–2019 by Statistics Austria [30] and in the first lockdown in Spring 2020 [6] there were differences between the samples in psychological quality of life score (WHOQOL-BREF, F (2, 7,420,382) = 607.6, *p* < 0.001), the mean PHQ-8 score (F (2, 7,420,382) = 1334, *p* < 0.001) and the likelihood of being over the cut-off for clinically relevant depression (ꭓ^2^ (2) = 1463, *p* < 0.001). Specifically, Bonferonni corrected post hoc tests revealed that the current sample (December/January lockdown) had a significantly lower quality of life score than the pre-pandemic sample (*p* < 0.001) but did not statistically differ from the first lockdown sample (*p* = 0.38). The current sample also had a significantly higher mean PHQ-8 score for depression than both the pre-pandemic sample (2018–2019 ATHIS sample, *p* < 0.001) and the sample from the April 2020 lockdown (*p* < 0.001). Further details can be found in Appendix A.

Finally, the percentage of people over the cut-off values for moderate depression, anxiety and insomnia from the December/January lockdown are presented alongside those from the sample surveyed in the April lockdown in Austria [6]. Overall, the percentage of people over the cut-offs of all measures have slightly increased in both genders but with no significant interaction between gender and lockdown (April vs. December/January, Table 7). There was a significant interaction between lockdown and age category on the likelihood of being over the depression cut-off (F (5, 2498) = 2.15, *p* = 0.05); the percentage of people over the depression cut-off has significantly increased between the two lockdowns in the youngest (t (269) = 3.03, *p* < 0.01) and oldest age groups (t (317 = 2.02, *p* < 0.05, Figure 1) suggesting the December/January lockdown has especially affected the depression levels of these groups. No other age groups showed a difference between the two lockdowns (Appendix A). There was no interaction between lockdown and age category on the likelihood of moderate anxiety symptoms (F (5, 2498) = 0.76, *p* > 0.05) nor moderate insomnia symptoms (F (5, 2498) = 0.94, *p* > 0.05). Finally, there was no interaction between lockdown and marital status on the likelihood of being over the cut-off for depression (F (5, 2498) = 0.88, *p* > 0.05)), anxiety, (F (5, 2498) = 0.76, *p* > 0.05) or insomnia (F (5, 2498) = 0.77, *p* > 0.05, see also Appendix A).

## 4. Discussion

Overall a notable decline in mental health in Austria across many measures has been seen over the 2020/2021 winter holiday period, as compared to both pre-pandemic levels [30] and the first lockdown in April 2020 [22]. In particular, those under 24 and over 65 years of age have shown a large increase in depressive symptoms compared to the April lockdown. Thus, as the pandemic continues, mental health appears to be deteriorating.

Studies conducted in previous years can put the current findings in context. The Austrian Health Survey [30], conducted before the pandemic, between October 2018 and September 2019, found 5.6% of the population to be over the cut-off for moderate depression, significantly lower than the 25.1% in the current sample. Similarly, other representative population pre-pandemic surveys show, for example, the percentage of people over the cut-off for moderate depression symptoms to be 6% in Germany in 2013 [46]. Our sample is however comparable to another study conducted during COVID-19 in China, which reports a 20% depression prevalence using another scale [47]. Similarly, a mean score of 3.6 was reported on the GAD-7 measure of anxiety in Germany in 2017 [48], compared to 6.25 in the current results obtained during a COVID-19 lockdown in Austria. Stress levels measured by the PSS-10 appear similar in the current study (mean 16.4) as in a large study of 41 countries during COVID-19 (mean 17.4) [49]. Regarding quality of life (WHOQOL) the ATHIS 2019 [30] reports mean WHOQOL Psychological scores of 81.11 for men and 78.43 for women, which is significantly higher than those reported by the current participants (71.87 and 66.0, respectively). These comparisons reinforce findings that mental health has drastically declined during the COVID-19 pandemic.

As well as a decline since pre-pandemic levels, our data from December/January also shows a further decline in mental health in Austria in comparison with the April lockdown that occurred in spring 2020. It has already been shown that the reduction in mental health observed in the first lockdown [6] remained despite the easing of restrictions over the summer [22] in Austria. The results of the current study demonstrate that with the introduction of another lockdown, over the Christmas holiday period, this reduced level of psychological well-being has been further affected. Furthermore, our findings corroborate those already published regarding the groups most vulnerable to psychological impact from the pandemic, specifically; women [13,17,18], young adults [13,14,15,16,17], those with financial concerns [6] and those who do not partake in any physical exercise [19].

The effect that women are more burdened than men was to be expected as this is a general trend in psychiatric research [50] and representative population surveys [30,51]. However, it should be noted that the relative change in some mental health measures from pre-COVID levels is higher for women than men. In particular, 15% more men are above the cut-off for depression in the current study compared to the ATHIS 2019 findings, whereas 24% more women are above this cut-off. Likewise, compared to a representative population survey in Germany in 2017 [48], 12.7% more men, but 21.9% more women are above the cut-off for anxiety in the current sample. Perceived stress has also increased more in women than men with 4.7 and 2.8 point increases, respectively, compared to a German survey in 2016 [52]. Quality of life, however, has not disproportionately worsened in women; compared to the pre-pandemic survey men have shown a 9-point decrease and women a 12-point decrease in their WHOQOL scores. Although the current results are not directly comparable with the pre-COVID studies, these disparities do indicate that the pandemic may be affecting women more than men. There are risk factors experienced by women which are likely intensified during a pandemic [53], for example, increased likelihood of unemployment, childcare and home keeping responsibilities [54], and intimate partner and gender-based violence (reviewed in: [51]). Therefore, there is a need to incorporate gender equality into any COVID-19 decision making.

We also observed an effect of age group on mental health. Of particular concern are young adults under the age of 24, whereby an alarming 50% are showing depression symptoms above the cut-off for clinical relevance and 34% show clinically relevant anxiety symptoms. These results reflect those of a study in Canada where 66% of under 25s scored above the same depression cut-off and 65% above the anxiety cut-off [55]. The finding that young people are showing poorer mental health than other age groups may at first glance appear counterintuitive as older adults are indeed physically more at risk from the virus. However, as Salari et al. [9] note, younger adults seem to have higher concern over the future consequences and are more affected by societal and economic challenges that this situation may bring. Furthermore, the inability to socialise as normal over the holiday period may have further exacerbated these effects in young people. Our sample consisted of adults and therefore given these findings, future research should also consider whether under 18s are also showing such a mental health decline as this would be an important group for targeting timely interventions.

Additionally, the over 65s, while showing better scores than younger people in general, have doubled in the percentage of people showing above cut-off depressive symptoms over the Christmas lockdown as compared to the first lockdown (11.8% vs. 5%). Given that Christmas can be especially important for reducing loneliness in older age groups [23], it is not surprising that the reduced social contact over this time has been damaging to the mental health of the over 65s. Further studies will be needed after the holiday period to assess whether this decline is specifically related to Christmas or if we are witnessing an overall decline in mental health in the youngest and oldest adults of the population. However, we do also see slightly higher levels of depressive symptoms in single and separated individuals as compared to the first lockdown, which we do not see in other family groups such as married people, suggesting that at least some of the effects seen in the current study are related to the lockdown occurring over a traditionally important family holiday. Indeed, the levels of depression symptoms changed very little in married and cohabiting couples between the April and December/January lockdowns, indicating that perhaps the ability to gather in, albeit small, groups on the 24 and 25 December may have had a buffering effect on the poor mental health in some people.

Altogether, the long-term and unpredictable nature of the pandemic is taking a toll on the nation’s mental health. Furthermore, having a lockdown over the holiday period has likely added a mental burden to some groups. It is encouraging to note that doing exercise as little as at least one day per week for 30 min or more provides somewhat of a buffer to the mental health challenges we are faced with. It is known that exercise promotes better mental health [56] and is a simple strategy that can be recommended among general populations.

Overall, these findings can have important implications for provisions of mental health aid. Depending on the severity of the symptoms as well as the age of the affected, different offers of help are required. First, and most important, the existing offers of support, such as psychiatric-, psychological-, and psychotherapeutic offers should be expanded. Covid-appropriate group therapeutic interventions, in particular, could be helpful. Furthermore, a large group could be reached with online interventions in a variety of settings. Especially for the less severe cases, health-promoting measures, such as physical exercise or sport, should be advertised more intensively. Psychosocial consequences of the pandemic should also be given greater consideration for the further decisions to contain the pandemic.

There are some limitations to the study that should be noted. Firstly, due to the time sensitive nature of the data collection, some sub-groups did not reach the desired quota for a representative sample (Appendix A), for example the high education group and young men. Related to this, some sub-groups, namely separated and widowed people and those with an income below €1000, were rather small meaning we must take caution in any conclusions drawn from results including these groups. Additionally, there is a dearth of studies on mental health immediately before the pandemic and therefore we are unable to conclude that the pandemic per se has caused the observed decline in mental health. Furthermore, because the pre-pandemic data was collected over the course of a year and the current data were collected specifically over the Christmas period, we are unable to discern which aspects of these findings are due to the long-term nature of the pandemic and which are specifically due to having a lockdown over an important social and cultural event. Additionally, we could not control for pre-existing mental health conditions, which may have affected some results. However, this is also the case for other large-scale representative surveys and therefore the comparisons are valid.

Lastly, as this was a large-scale online survey, all of the measures are self-ratings rather than clinician-based assessments. This allowed a rapid assessment without direct social contact but it is known that people can be biased when reporting on their own experiences [57] and therefore structured clinical interviews would provide a more objective measure.

## 5. Conclusions

The COVID-19 pandemic has resulted in a mental health decline around the world. The current results show that in Austria, this trend is worsening with further lockdown, occurring over the Christmas period. Certain sociodemographic and lifestyle-related factors such as exercise, being married and having a higher income can buffer these effects. However, psychological support, particularly for those groups showing the highest burden on their mental health such as young people, women and people with low incomes, should be offered to slow down mental health deterioration.

## Figures and Tables

**Figure 1 ijerph-18-03679-f001:**
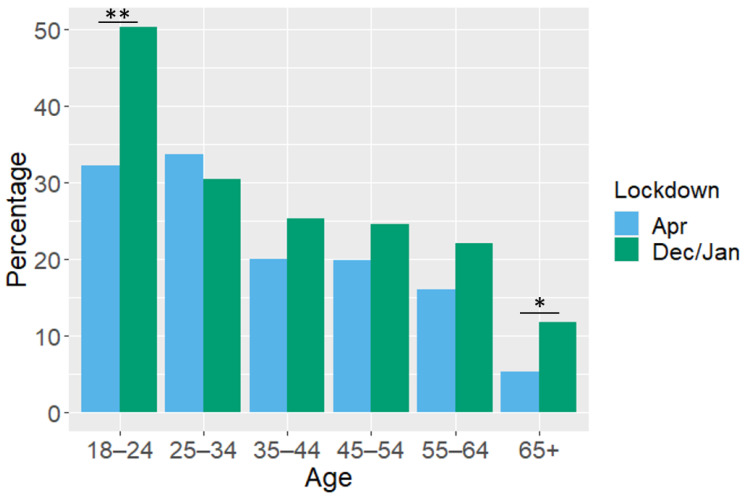
Percentage of people over the cut-off (≥10) for moderate depressive symptoms on the PHQ-9 in the April and December/January lockdowns according to age category. ** <0.01. * <0.05.

**Table 1 ijerph-18-03679-t001:** Study sample characteristics (*N* = 1505).

Variable	*N*	%
Total	1505	
Gender		
Women	764	49.24
Men	741	50.76
Other	0	0
Age		
18–24	153	10.17
25–34	279	18.54
35–44	289	19.2
45–54	326	21.66
55–64	272	18.07
65+	186	12.36
Region		
Burgenland	58	3.85
Lower Austria	305	20.27
Vienna	346	22.99
Carinthia	103	6.84
Styria	221	14.68
Upper Austria	227	15.08
Salzburg	79	5.25
Tyrol	114	7.57
Vorarlberg	52	3.46
Education		
Low	40	2.66
Middle	809	53.75
High	656	43.59

**Table 2 ijerph-18-03679-t002:** Number of people above and below the cut-offs for depression, anxiety and insomnia and mean scores for measures of psychological health and wellbeing for the total sample and by gender.

		Gender			
		Male	Female	Total	Statistic
PHQ-9 score *N* (%)	<10	585 (78.95)	523 (68.46)	1108 (73.62)	χ^2^ (1) = 21.32, *p* < 0.001
	≥10	156 (21.05)	241 (31.54)	397 (26.38)	
GAD-7 score *N* (%)	<10	616 (83.13)	540 (70.68)	1156 (76.81)	χ^2^ (1) = 32.74, *p* < 0.001
	≥10	125 (16.87)	224 (29.32)	349 (23.19)	
ISI score *N* (%)	<15	620 (83.67)	604 (79.06)	1224 (83.33)	χ^2^(1) = 5.27, *p* < 0.05
	≥15	121 (16.33)	160 (20.94)	281 (18.67)	
	Total (*N*)	741	764	1505	
PHQ-9	Mean (SD)	5.87 (5.41)	7.69 (6.11)	6.8 (5.84)	t (1503) = −6.11, *p* < 0.001, g = 0.32
GAD-7	Mean (SD)	5.34 (4.73)	7.12 (5.24)	6.25 (5.07)	t (1503) = −7.0, *p* < 0.001, g = 0.36
ISI	Mean (SD)	8.42 (6.0)	9.5 (6.15)	8.97 (6.1)	t (1503) = −3.46, *p* < 0.001, g = 0.18
WHOQOL BREF (psychological domain)	Mean (SD)	71.87 (17.64)	66.0 (19.16)	68.89 (18.66)	t (1503) = −6.19, *p* < 0.001, g = 0.32
WHO-5	Mean (SD)	15.09 (5.56)	13.6 (5.92)	14.34 (5.79)	t (1503) = 5.03, *p* < 0.001, g = 0.26
PSS-10	Mean (SD)	15.03 (7.21)	17.78 (7.73)	16.42 (7.6)	t (1503) = −7.14, *p* < 0.001, g = 0.37

*p*: *p*-values (2-tailed); *N*: frequencies; SD: standard deviation, χ^2^: Chi-square; t: *t*-test; ISI: Insomnia Severity Index, GAD-7; Generalized Anxiety Disorder 7 scale; PHQ-9: Patient Health Questionnaire 9 scale; WHO-QOL BREF: Quality of Life questionnaire of the World Health Organization (WHO); WHO-5: Well-being questionnaire of the World Health Organization (WHO); PSS-10: Perceived Stress Scale 10.

**Table 3 ijerph-18-03679-t003:** Number of people above and below the cut-offs for depression, anxiety and insomnia and mean scores for measures of psychological health and wellbeing by age group.

Age (years)								
		18–24	25–34	35–44	45–54	55–64	65+	Statistic
PHQ-9 score *N* (%)	<10	76 (49.67)	194 (69.53)	216 (74.74)	246 (75.46)	212 (77.94)	164 (88.17)	χ^2^ (5) = 71.23, *p* < 0.001
	≥10	77 (50.33)	85 (30.47)	73 (25.26)	80 (24.54)	60 (22.06)	22 (11.83)	
GAD-7 score *N* (%)	<10	100 (65.36)	200 (71.68)	222 (76.82)	246 (75.15)	221 (81.25)	167 (89.78)	χ^2^ (5) = 36.3, *p* < 0.001
	≥10	53 (34.64)	79 (28.32)	67 (23.18)	80 (24.85)	51 (18.75)	19 (10.22)	
ISI score *N* (%)	<15	114 (74.51)	228 (81.72)	229 (79.24)	257 (78.83)	231 (84.93)	165 (88.71)	χ^2^ (5) = 15.87, *p* < 0.01
	≥15	39 (25.49)	51 (18.28)	60 (20.76)	69 (21.17)	41 (15.07)	21 (11.29)	
	Total (*N*)	153	279	289	326	272	186	
PHQ-9	Mean (SD)	9.92 (5.98)	7.89 (5.88)	6.91 (5.71)	6.6 (6.04)	5.87 (5.6)	4.12 (4.18)	F (5, 1499) = 21.26, *p* < 0.001, η^2^ = 0.027
GAD-7	Mean (SD)	8.42 (5.27)	7.08 (4.72)	6.4 (4.86)	6.31 (5.44)	5.5 (5.08)	3.95 (3.89)	F (5, 1499) = 16.85, *p* < 0.001, η^2^ = 0.053
ISI	Mean (SD)	10.42 (5.95)	9.54 (5.83)	9.37 (6.07)	9.28 (6.48)	8.21 (6.1)	6.85 (5.34)	F (5, 1499) = 8.15, *p* < 0.001, η^2^ = 0.026
WHOQOL BREF (psychological domain)	Mean (SD)	63.13 (17.96)	66.02 (19.51)	69.26 (16.63)	68.92 (19.96)	70.88 (19.38)	74.4 (15.36)	F (5, 1499) = 8.31, *p* < 0.001, η^2^ = 0.027
WHO-5	Mean (SD)	12.33 (5.29)	13.79 (5.32)	14.04 (5.76)	14.08 (6.19)	15.06 (6.09)	16.66 (4.86)	F (5, 1499) = 11.69, *p* < 0.001, η^2^ = 0.038
PSS-10	Mean (SD)	20.01 (7.1)	17.98 (7.1)	17.03 (7.01)	16.14 (8.11)	15.0 (7.84)	12.77 (6.26)	F (5, 1499) = 21.5, *p* < 0.001, η^2^ = 0.067

*p*: *p*-values (2-tailed); N: frequencies; SD: standard deviation, χ^2^: Chi-square; t: *t*-test; ISI: Insomnia Severity Index, GAD-7: Generalized Anxiety Disorder 7 scale; PHQ-9: Patient Health Questionnaire 9 scale; WHO-QOL BREF: Quality of Life questionnaire of the World Health Organization (WHO); WHO-5: Well-being questionnaire of the World Health Organization (WHO); PSS-10: Perceived Stress Scale 10.

**Table 4 ijerph-18-03679-t004:** Number of people above and below the cut-offs for depression, anxiety and insomnia and mean scores for measures of psychological health and wellbeing by marital status.

Marital Status								
		Single	Separated	Divorced	Cohabiting	Married	Widowed	Statistic
PHQ-9 score *N* (%)	<10	283 (64.91)	23 (53.49)	73 (69.52)	245 (75.15)	457 (81.46)	27 (79.41)	χ^2^ (5) = 45.67, *p* < 0.001
	≥10	153 (35.09)	20 (46.51)	32 (30.48)	81 (24.85)	104 (18.54)	7 (20.59)	
GAD-7 score *N* (%)	<10	318 (72.94)	28 (65.12)	80 (76.19)	247 (75.77)	455 (81.11)	28 (82.35)	χ^2^ (5) = 13.59, *p* < 0.05
	≥10	118 (27.06)	15 (34.88)	25 (23.81)	79 (24.23)	106 (18.89)	6 (17.65)	
ISI score *N* (%)	<15	346 (79.36)	27 (62.79)	83 (79.05)	271 (83.13)	468 (83.42)	29 (85.29)	χ^2^ (5) = 13.87, *p* < 0.05
	≥15	90 (20.64)	16 (37.21)	22 (20.95)	55 (16.87)	93 (16.58)	5 (14.71)	
	Total (*N*)	436	43	105	326	561	34	
PHQ-9	Mean (SD)	8.09 (6.29)	8.86 (6.46)	6.87 (5.96)	6.86 (5.36)	5.66 (5.44)	5.44 (5.92)	F (5, 1499) = 10.27, *p* < 0.001, η^2^ = 0.033
GAD-7	Mean (SD)	6.9 (5.25)	7.98 (5.0)	6.2 (5.34)	6.52 (4.87)	5.54 (4.91)	4.76 (4.86)	F (5, 1499) = 5.44, *p* < 0.001, η^2^ = 0.018
ISI	Mean (SD)	9.37 (6.06)	11.44 (6.28)	9.43 (6.65)	8.83 (5.81)	8.52 (6.14)	7.97 (5.75)	F (5, 1499) = 2.76, *p* < 0.05, η^2^ = 0.009
WHOQOL BREF (psychological domain)	Mean (SD)	64.57 (19.71)	61.82 (19.35)	68.45 (17.92)	68.9 (18.05)	72.55 (17.55)	74.02 (15.89)	F (5, 1499) = 11.1, *p* < 0.001, η^2^ = 0.036
WHO-5	Mean (SD)	13.27 (6.05)	12.26 (6.23)	14.22 (6.18)	14.24 (5.41)	15.27 (5.55)	16.53 (5.06)	F (5, 1499) = 8.19, *p* < 0.001, η^2^ = 0.027
PSS-10	Mean (SD)	17.5 (7.89)	20.49 (7.67)	15.13 (7.8)	17.01 (7.27)	15.36 (7.24)	13.32 (7.8)	F (5, 1499) = 8.74, *p* < 0.001, η^2^ = 0.028

*p*: *p*-values (2-tailed); *N*: frequencies; SD: standard deviation, χ^2^: Chi-square; t: *t*-test; ISI: Insomnia Severity Index, GAD-7: Generalized Anxiety Disorder 7 scale; PHQ-9: Patient Health Questionnaire 9 scale; WHO-QOL BREF: Quality of Life questionnaire of the World Health Organization (WHO); WHO-5: Well-being questionnaire of the World Health Organization (WHO); PSS-10: Perceived Stress Scale 10.

**Table 5 ijerph-18-03679-t005:** Number of people above and below the cut-offs for depression, anxiety and insomnia and mean scores for measures of psychological health and wellbeing by income.

Income (Monthly, €)							
		<1000	1000–2000	2000–3000	3000–4000	>4000	Statistic
PHQ-9 score *N* (%)	<10	73 (56.59)	212 (66.04)	300 (73.89)	244 (76.01)	279 (85.06)	χ^2^ (4) = 51.82, *p* < 0.001
	≥10	56 (43.41)	109 (33.96)	106 (26.11)	77 (23.99)	49 (14.94)	
GAD-7 score *N* (%)	<10	81 (62.79)	230 (71.65)	312 (76.85)	250 (77.88)	283 (86.28)	χ^2^ (4) = 35.75, *p* < 0.001
	≥10	48 (37.21)	91 (28.35)	94 (23.15)	71 (22.12)	45 (13.72)	
ISI score *N* (%)	<15	90 (69.77)	250 (77.88)	333 (82.02)	259 (80.69)	292 (89.02	χ^2^ (4) = 26.87, *p* < 0.01
	≥15	39 (30.23)	71 (22.12)	73 (17.98)	62 (19.31)	36 (10.98)	
	Total (*N*)	129	321	406	321	328	
PHQ-9	Mean (SD)	9.49 (7.28)	7.73 (6.33)	6.83 (5.46)	6.36 (5.59)	5.2 (4.82)	F (4, 1500) = 16.03, *p* < 0.001, η^2^ = 0.041
GAD-7	Mean (SD)	8.02 (5.84)	6.89 (5.39)	6.29 (4.78)	6.07 (4.99)	5.03 (4.54)	F (4, 1500) = 10.3, *p* < 0.001, η^2^ = 0.027
ISI	Mean (SD)	10.67 (6.49)	9.86 (6.19)	9.14 (6.0)	8.92 (6.17)	7.27 (5.51)	F (4, 1500) = 10.93, *p* < 0.001, η^2^ = 0.028
WHOQOL BREF (psychological domain)	Mean (SD)	59.95 (21.61)	65.1 (19.37)	68.08 (17.65)	70.9 (17.82)	75.15 (16.26)	F (4, 1500) = 22.28, *p* < 0.001, η^2^ = 0.056
WHO-5	Mean (SD)	12.32 (6.86)	13.56 (6.0)	14.31 (5.48)	14.57 (5.56)	15.69 (5.39)	F (4, 1500) = 11.69, *p* < 0.001, η^2^ = 0.038
PSS-10	Mean (SD)	18.95 (9.05)	17.65 (7.6)	16.53 (7.24)	15.91 (7.45)	14.6 (7.1)	F (4, 1500) = 11.05, *p* < 0.001, η^2^ = 0.029

*p*: *p*-values (2-tailed); *N*: frequencies; SD: standard deviation, χ^2^: Chi-square; t: *t*-test; ISI: Insomnia Severity Index, GAD-7: Generalized Anxiety Disorder 7 scale; PHQ-9: Patient Health Questionnaire 9 scale; WHO-QOL BREF: Quality of Life questionnaire of the World Health Organization (WHO); WHO-5: Well-being questionnaire of the World Health Organization (WHO); PSS-10: Perceived Stress Scale 10.

**Table 6 ijerph-18-03679-t006:** Number of people above and below the cut-offs for depression, anxiety and insomnia and mean scores for measures of psychological health and wellbeing by days of physical activity (at least 30 min).

Physical Activity (Days/Week)							
		0	1	2	3	>4	Statistic
PHQ-9 score *N* (%)	<10	135 (61.36)	135 (72.58)	197 (71.64)	205 (75.65)	436 (78.84)	χ^2^ (4) = 26.02, *p* < 0.001
	≥10	85 (38.64)	51 (27.42)	78 (28.36)	66 (24.35)	117 (21.16)	
GAD-7 score *N* (%)	<10	151 (68.64)	143 (76.88)	209 (76)	222 (81.92)	431 (77.94)	χ^2^ (4) = 12.72, *p* < 0.05
	≥10	69 (31.36)	43 (23.12)	66 (24)	49 (18.08)	122 (22.06)	
ISI score *N* (%)	<15	162 (73.64)	146 (78.49)	230 (83.64)	231 (85.24)	455 (82.28)	χ^2^ (4) = 13.58, *p* < 0.01
	≥15	58 (26.36)	40 (21.51)	45 (16.36)	40 (14.76)	98 (17.72)	
	Total (*N*)	220	186	275	271	553	
PHQ-9	Mean (SD)	8.66 (6.39)	7.21 (6.0)	7.16 (5.4)	6.48 (5.38)	5.87 (5.81)	F (4, 1500) = 10.08, *p* < 0.001, η^2^ = 0.026
GAD-7	Mean (SD)	7.43 (5.65)	6.56 (5.3)	6.29 (4.53)	5.92 (4.63)	5.81 (5.15)	F(4, 1500) = 4.52, *p* < 0.001, η^2^ = 0.012
ISI	Mean (SD)	10.1 (6.01)	8.89 (6.22)	9.09 (5.54)	8.64 (5.86)	8.64 (6.42)	F(4, 1500) = 2.53, *p* < 0.05, η^2^ = 0.007
WHOQOL BREF (psychological domain)	Mean (SD)	60.64 (20.5)	67.29 (19.27)	68.32 (15.83)	69.74 (17.23)	72.57 (18.59)	F(4, 1500) = 17.41, *p* < 0.001, η^2^ = 0.044
WHO-5	Mean (SD)	11.66 (6.13)	13.54 (5.6)	14.11 (5.11)	14.77 (5.5)	15.57 (5.79)	F(4, 1500) = 20.35, *p* < 0.001, η^2^ = 0.051
PSS-10	Mean (SD)	19.04 (7.96)	17.05 (7.82)	16.67 (6.67)	15.8 (7.24)	15.36 (7.73)	F(4, 1500) = 10.3, *p* < 0.001, η^2^ = 0.027

*p*: *p*-values (2-tailed); *N*: frequencies; SD: standard deviation, χ^2^: Chi-square; t: *t*-test; ISI: Insomnia Severity Index, GAD-7: Generalized Anxiety Disorder 7 scale; PHQ-9: Patient Health Questionnaire 9 scale; WHO-QOL BREF: Quality of Life questionnaire of the World Health Organization (WHO); WHO-5: Well-being questionnaire of the World Health Organization (WHO); PSS-10: Perceived Stress Scale 10.

**Table 7 ijerph-18-03679-t007:** Percentage of people over the cut-offs for moderate depression, anxiety and insomnia by gender in the April and December/January lockdowns.

		April Lockdown		December/January Lockdown		Statistic
		Male	Female	Male	Female	
PHQ-9 score (%)	≥10	16.4	25.1	21.1	31.5	Lockdown: t (2506) = 1.85, *p* = 0.06 Lockdown * gender: t (2506) = 0.53, *p* > 0.05
GAD-7 score (%)	≥10	14.1	23.4	16.9	29.3	Lockdown: t(2506) = 1.16, *p* > 0.05 Lockdown * gender: t (2506) = 0.95, *p* > 0.05
ISI score (%)	≥15	14.3	17.0	16.3	20.9	Lockdown: t(2506) = 0.9, *p* > 0.05 Lockdown * gender: t (2506) = 0.63, *p* = 0.6

PHQ-9: Patient Health Questionnaire 9 scale, GAD-7: Generalized Anxiety Disorder 7 scale, ISI: Insomnia Severity Index, * interaction term

## Data Availability

Data is available upon reasonable request.

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
