# Peer review of "Mental Health during the COVID-19 Lockdown over the Christmas Period in Austria and the Effects of Sociodemographic and Lifestyle Factors"

_ijerph, 2021, doi:10.3390/ijerph18073679_

Round 1
Reviewer 1 Report
This paper describes exploratory analyses to investigate mental health and wellbeing in a population representative sample aged 18-65+ (N=1505) in Austria, during the Christmas lockdown period, and to determine the sociodemographic and lifestyle factors associated with individual differences in the outcome measures.
The authors provide a good rationale for this study and the paper is well-written and concise. The main findings are clearly communicated, which can be difficult in studies such as these, which have multiple health outcomes. Comparing the current results with those of pre-pandemic studies of mental health in the Austrian population provides a useful narrative and helps to illustrate the declining mental health of the nation as a result of Covid-19 restrictions. Nevertheless, there are a few issues detailed below which should be addressed, the first of which is imperative before the paper can be properly reviewed.
- There is no covariates/variables section. This is an important omission and leaves the reader clueless as to how these variables were calculated and coded, and therefore whether their inclusion in the models were statistically correct. Firstly, how was physical activity ascertained, in what unit was it measured, and how was it coded? Does “1”=30 mins a week, “2”=60mins a week etc? The Supplementary Materials table S5 suggest that the numbers represent ‘days’ of physical activity, but either way, it is very unclear. Secondly, in what unit is income measured? Thirdly, exactly how was marital status coded in the models? Unlike the other variables (e.g. age, income), it has no ordinal scale. That is, single cannot be coded as 1, separated as 2, divorced as 3 etc., as to do so would be statistically incorrect. Also I think the term ‘family status’ is slightly misleading as it usually refers to living with other family members rather than just marital status. Please also provide units of measurement in all of the tables either in footnotes or the table itself.
The Statistical Analysis section (line 158) states that “Descriptive statistics describe the demographic characteristics of this representative Austrian sample, mean values of each measure, and the percentage of each sub-group (gender, age, family status, income and physical activity)” but this data does not appear in the manuscript? Please add in income, marital status and physical activity to the variable list in Table 1 so that *all* the data is presented here in the correct place.
- The Introduction provides a good rationale for studying mental health over the Christmas period and indicates why negative mental health is likely to be amplified at this time, given the restrictions and that this period can be a stressor anyway. However, this point is not necessarily carried through to the Discussion. For example, in the para beginning line 283, the authors should add that comparisons between the current results and those of pre-pandemic MH levels might be problematic because the previous cited study spans a significantly longer time period (Oct 2018-Sep 2019) and therefore is more representative of general MH than the current study which has taken place not only over an extremely short time period, but at a time when MH issues are likely to be artificially inflated. I would suggest some modification of the Discussion, including the Limitations section, to this effect.
- Related to the above, it could be argued that the lack of socialising/attending parties etc over Christmas lockdown is likely to disproportionately affect younger people, and therefore the lower MH scores of this age group in this 2-week window, may not necessarily reflect habitual levels.
- There are a few additional limitations which have not been included in the Discussion. Given that this is an online survey, taking part be biased towards those with a certain degree of technological proficiency, especially in the 65+ age group. Secondly, the authors did not ask about clinical diagnoses of depression, anxiety etc, and were therefore unable to control for existing (not pandemic-related) MH conditions.
- There are a few minor points:
- line 36. Explain “drastic changes to our lifestyles” for the reader.
- line 44-45. This is just a suggestion, but would it not be better here to say “identify those most at risk…” rather than “identify sociodemographic and lifestyle-related risk factors to mental health…”?
- line 158 – add a reference for R
- Table headings include the phrase “above the cut-offs” but it presents data for those below and above the cut-offs. Please edit accordingly.
- in general, the tables could be presented more neatly.
Reviewer 2 Report
- It is not clear why the author chose to use low, middle, and high as the category of education level. Why not report the more standard/objective categories, i.e., primary school, secondary school, etc.
- In the first paragraph of the result section, it maybe helpful to describe the general population of Austria. At least mention the population size of Austria in 2020 and what proportion the 1500 samples account for the total population. Did the author compare the characteristics of their sample with the government census data to check if their sample is representative?
- The authors should make it clearer that a general decline in mental health was observed between the two lockdowns in Austria. And maybe elaborate on the effect of prolong lockdown on mental health.
- Regarding the “Christmas lockdown”, the authors may elaborate on how loneliness maybe harmful to mental health.
- Lastly, except for purely describing the situation, the authors may consider what Austrian government could do to provide assistance to individuals in need regarding depressive symptoms, anxiety, etc.
